# Multivariate ERP Analysis of Neural Activations Underlying Processing of Aesthetically Manipulated Self-Face

**Hirokazu Doi** [1,2]

1 School of Science and Engineering, Kokushikan University, 4-28-1 Setagaya, Setagaya-ku, Tokyo 154-8515, Japan; hdoi_0407@yahoo.co.jp

2 Graduate School of Biomedical Sciences, Nagasaki University, Nagasaki 852-8520, Japan

**Abstract:** Representation of self-face is vulnerable to cognitive bias, and consequently, people often possess a distorted image of self-face. The present study sought to investigate the neural mechanism underlying distortion of self-face representation by measuring event-related potentials (ERPs) elicited by actual, aesthetically enhanced, and degraded images of self-face. In addition to conventional analysis of ERP amplitude and global field power, multivariate analysis based on machine learning of single trial data were integrated into the ERP analysis. The multivariate analysis revealed differential pattern of scalp ERPs at a long latency range to self and other familiar faces when they were original or aesthetically degraded. The analyses of ERP amplitude and global field power failed to find any effects of experimental manipulation during long-latency range. The present results indicate the susceptibility of neural correlates of self-face representation to aesthetical manipulation and the usefulness of the machine learning approach in clarifying the neural mechanism underlying self-face processing.

**Keywords:** face; self; ERP; multivariate analysis; temporal generalization

## 1. Introduction

A subjective evaluation of one's attributes colours their psychological landscape and could be considered one of the primary determinants of their life course [1,2]. Existing studies on self-recognition have repeatedly shown self-serving bias in the evaluation of one's attributes [3]. The most well-recognized illustration of this phenomena is "the Lake Wobegon effect" according to which the majority of people estimate their ability "above average" in several domains such as understanding of humour, driving skills, intellectual ability, and socially desirable personality traits [4,5]. When feedback about one's ability and traits is delivered, positive appraisals are more likely to be integrated into self-evaluation than negative appraisals, which makes self-serving bias even stronger [3].

The mental representation of self-face and body is an integral constituent of self-concept [1,6]. Several studies have shown the possibility that even the representation itself, not the evaluation, of self-face and body could be deformed by cognitive bias. In the field of clinical psychology, people with body dysmorphia are known to have a deformed mental image of their body shape [7]. Epley et al. revealed that people show self-serving bias in representation of self-face [8]. In their study, a continuum of facial images was created by morphing participants' face and attractive or unattractive unfamiliar faces. When participants were prompted to select their face from the continuum, they were more likely to choose a morphed image falling between their face and an attractive one that is not their actual face.

These findings indicate that representation of self-appearance is not as stable as presumed and is malleable to psychological influences such as self-serving bias. Despite this, a majority of previous studies on cognitive and neural mechanism of self-face processing have used only actual self-face images as the stimuli [9–13], and a relatively small number of

studies have investigated the behavioural and neural responses to deformed representation of self-face that is different from actual self-face [14,15].

The primary goal of the present study was to fill in this gap in knowledge by measuring electrophysiological responses to actual and deformed images of self and other familiar faces. The experiment measured event-related potentials (ERPs) in response to three types of facial images of one's self and other familiar faces belonging to the same sex. Other familiar faces were used instead of complete stranger's faces to mitigate the influences of perceptual familiarity on electrophysiological responses. Only females were recruited as participants because previous studies have pointed out that females tend to be self-conscious of their physical appearance [16,17]. Three types of images included actual face, and two types of images comprised deformed face. One type of deformed image was created by exaggerating morphological features generally perceived to be attractive and the other type by diminishing them [18]. Thus, the three types of images can be deemed as actual, aesthetically enhanced, and degraded versions of self and other familiar face.

In conventional ERP studies, effects of experimental manipulation are assessed on mean amplitude of each ERP component independently. Many of the previous ERP studies on self-face recognition focused on P100, N170, and P250 [9,13,19,20]. Among these, P100 and N170 have been proposed to reflect low-level visual processing in primary visual cortex and face-selective activation in the occipito-temporal region [9,13,19,20], while P250 has been proposed to reflect the stage linked to face familiarity processing [19,20]. Thus, the effect of facial identity and aesthetical manipulation was investigated in these ERP components.

In addition to the conventional analysis, various methods for ERP and EEG analysis have been developed; fractional latency/amplitude measurement [21] aims to increase robustness of ERP measurement against fluctuating noise, and application of principal component analysis to ERP time series [22,23] to disentangle overlapping components. Another type of approach tried to detect ERP responses sensitive to experimental manipulation without a priori specification of ERP components and latency range of interest. These include ERP spatio-temporal cluster-based permutation of scalp ERP field [24] and microstate segmentation analysis [25]. A recent surge of interest in machine learning led to the utilization of machine learning algorithms in ERP/EEG analysis. This line of research mainly focused on prediction of internal states [26–29]. Besides a practical application, the machine learning approach had also given novel insights into the neurophysiological mechanism of perceptual and cognitive processing [30,31].

The present study adopted a state-of-the-art multivariate analysis to evaluate conditional differences in ERP based on machine learning of single-trial data [32,33]. In multivariate analysis of ERP, a classifier is trained to discriminate trials in one experimental condition from those in other experimental condition based on amplitude data from all the recorded channels at each time point. Multivariate analysis of ERP is largely data driven. Thus, in contrast to a conventional analysis of ERP peaks in which analyses are carried out for pre-chosen ERP components, multivariate analysis has the potential to find conditional differences that have hitherto been missed in the existing studies [33,34]. Multivariate analysis had first gained popularity in fMRI studies on object recognition [35,36]. The number of ERP studies adopting a multivariate approach is relatively small; however, several recent ERP studies have adopted this approach to further investigate the developmental course of face processing [37], face processing in pathological conditions [38], and the process of face memory formation [39].

## 2. Method

### 2.1. Participants

Ten pairs of right-handed females who had known each other for at least two years participated in the present study after giving written informed consent. Data from one pair were discarded due to failure in data storage. Thus, data from the remaining nine pairs (in total of 18 participants; M = 23.4 yrs old; SD = 4.9) were used for further analysis. They all

had normal or corrected-to-normal visual acuity. None reported history of psychiatric or neurological conditions. The protocol of this study was approved by the ethical committee of graduate school of biomedical sciences, Nagasaki University.

*2.2. Stimulus*

An image of each participant was taken against a cream-white background. The image was cropped so that the size of the image was 512 × 512 pixels. Based on this original image, aesthetically enhanced and degraded versions were created by exaggerating or diminishing the morphological features that are generally perceived to be attractive [18].

To achieve this, the shape of the original face was quantified by measuring the coordinates of 83 feature points of the original face. Based on the coordinates, the coordinates of the deformed face were calculated by a custom-made program. The coordinates of aesthetically enhanced versions of the faces were calculated by enlarging the eye region, shrinking the nose, and narrowing the contour of the lower face. In contrast to this, the coordinates of the aesthetically degraded versions were calculated by shrinking the eye region, enlarging the nose, and broadening the contour of the lower face. Examples of original, enhanced and degraded versions of face images are shown in Figure 1.

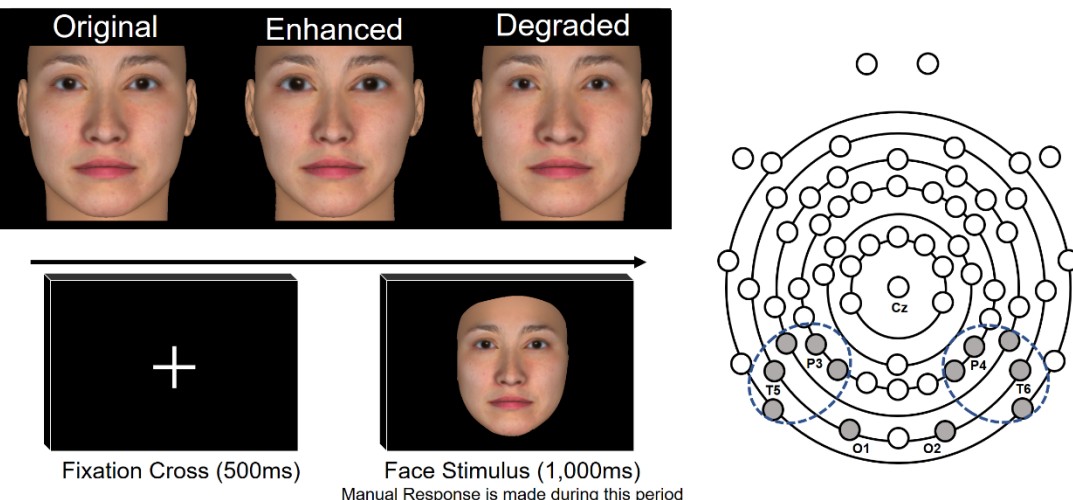

**Figure 1.** (**Left Upper panel**): Examples of the actual, aesthetically enhanced and degraded images of facial stimuli. These images were created using FaceGen software (Singular Inversions Inc. Tronto, Canada) for demonstration purposes only, and were not used in the actual experiment. (**Left Lower Panel**): Schematic representation of the temporal sequence of a single trial. (**Right Panel**): Representation of channel locations. The sensors used for ERP mean amplitude analyses are filled in grey. Sensors included in each occipito-temporal electrode cluster are circled by blue broken lines.

To synthesize the enhanced and degraded versions of the original face, the texture of the original face was warped so that the location of each feature point matched the location of the corresponding feature point of the aesthetically enhanced and degraded versions. These images were cropped in oval-like shape and were presented against a black background in a left–right reversed orientation following the previous studies on self-face recognition [40,41].

The stimuli were presented on a 19-inch display viewed from about 110 cm away. We wrote our experiments in Matlab, using the Psychophysics Toolbox extensions [42]. At the start of each trial, a fixation cross subtending 3.6 deg in height and 3.6 deg in width appeared at the centre of the screen for 500 ms. After the disappearance of the fixation cross, a stimulus face subtending about 7.8 deg in width and 7.8 in height was presented for 1 s. The temporal sequence of stimulus presentation is schematically shown in Figure 1. The faces comprised original, aesthetically enhanced, and degraded versions of self and other familiar faces. Participants viewed the paired participant's face in familiar condition.

Thus, participants were shown: Identity (2; Self, Familiar) × Version (3; Original, Enhanced, Degraded) = six types of faces.

### 2.3. Procedure

#### 2.3.1. EEG Measurement

After arrival at the lab, an EEG sensor net was placed on the participant's scalp. The experiment started after the preparation of the EEG recording. Participants were instructed to discriminate whether the presented face was her own face or not as soon as possible while facial stimuli stayed on the display. Participants made their responses by key-pressing with the left or right thumb. The correspondence between key and response (self or other's face) was counterbalanced across pairs. If they did not answer within the period of face stimulus presentation, response in the trial was treated as incorrect. They were told to classify aesthetically enhanced, aesthetically degraded and original versions of self-face as their own face.

Each of the six types of faces were presented 90 times throughout the experiment, resulting in a total of 540 experimental trials. The trials of six conditions were administered in a pseudorandomized order. Pseudo-randomization of trials was achieved by the following procedure. First, trials of the six conditions were randomized to form a sub-cluster of six trials. Each block was created by concatenating 15 sub-clusters of trials. Thus, the number of trials in each of the six conditions was equated to be fifteen within each block. Complete randomization of 540 trials was not adopted to avoid the situation that trials of identical trials are repeated many times in succession. The experimental trials were separated into six experimental blocks with a brief rest for refreshments. Each block lasted for about 2.3 min. The length of brief rest varied across participants, but was generally shorter than 1.5 min. The rest period after the third block lasted for about 5 min for checking and, if necessary, lowering contact impedance. EEG referenced to Cz was recorded by Geodesic 64 ch EEG System (Electrical Geodesics Inc., Eugene, OR, USA) in 1kHz and stored in hard disk.

#### 2.3.2. Post ERP Measurement Experiment

The behavioural task was administered after the EEG experiment to measure participants' subjective evaluations of the six types of faces. At each trial, one of the six faces was presented on the left side of the display. Simultaneously, three vertical trackbars with 21 tics were presented to the right side of the face. The upper and lower edges of these trackbars were labelled "Attractive–Unattractive", "Pretty–Not Pretty", and "Similar to me–Not similar to me", respectively. "Pretty–Not Pretty" evaluation was obtained because "*Kawaii*", the Japanese word for "Pretty", is used to express many aspects of favourable impression [43]. Thus, measurement of "Pretty–Not Pretty" evaluation is expected to reveal aesthetic evaluation not captured by "Attractive–Unattractive" dimension. The participants evaluated the presented face by moving the trackbars to the location closest to their subjective evaluation. After moving the trackbars to the preferred location, participants clicked the "proceed" button directly below the trackbars. Clicking the button started the next trial. Each of the Identity (2) × Version (3) = six types of faces were evaluated twice, resulting in 12 experimental trials. The order of stimulus presentation was pseudo-randomly determined.

### 2.4. Analysis

#### 2.4.1. Behaviour

The reaction time (RT) data of correct trials in EEG measurement were entered into an analysis of variance (ANOVA) with within-participant factors of Identity (2) and Version (3). Subjective evaluations of perceived attractiveness, prettiness, and self-similarity were analysed by ANOVAs with the same factorial design. The significance threshold was set to 0.05. Statistical analyses were carried out by R Analytic Flow (ef-prime Inc., Tokyo, Japan) and anovakun (retrieved from http://riseki.php.xdomain.jp/index.php?ANOVA%

E5%90%9B, accessed on 10 July 2021). Significance threshold of multiple comparison was adjusted by Modified Sequentially Rejective Bonferroni procedure.

2.4.2. ERP

EEG data were pre-processed using EEGLab [44]. EEG data were first downsampled to 250 Hz (each time point covering 4 ms), bandpass filtered (0.1–30 Hz) and re-referenced to average reference. The data from two electrodes (E62, E63) on the facial surface mainly for the detection of eye-movement artifacts and those near the tragus (E23, E55) were deleted from the dataset, and the data from the remaining 61 channels were entered into further analysis. The channel layout on the scalp surface is shown in Figure 1.

Artifacts were removed from the data by performing independent component (IC) analysis. After decomposition, ICs were checked by visual inspection, and those judged to reflect blink and motion artifacts based on scalp distribution and temporal fluctuation were removed from the data, resulting in removal of on average 3.9 ICs. EEG data were epoched to $-100$–700 ms after stimulus onset and baseline-corrected with $-100$ to 0 ms as the baseline. Because the data were downsampled to 250 Hz, each epoch contained 200 time points. The trials in which EEG exceeded $\pm 75$ µV were excluded from further analysis and visually checked afterwards. After pre-processing, 69.7 trials on average were retained for further analyses. In self condition, 69.4, 71.0, 70.8 trials were retained for original, enhanced and degraded versions, whereas in familiar condition, 67.8, 69.1, and 70.3 trials for original, enhanced and degraded versions, respectively.

Conventional ERP Analysis

ERP was computed by averaging the epochs of the same condition. All the eligible trials were used for computation of ERP to increase the signal-to-noise ratio. Previous studies revealed the effect of facial familiarity on early visual components in left and right occipito-temporal regions [9,13,19,20]. The mean amplitude of P100 was measured in bilateral occipital sensors (E39 for the right and E35 for the left) within 100 to 140 ms after stimulus onset.

The left and right occipito-temporal clusters each included five electrode locations (E40, E42, E45, E44, E43 for the right cluster, and E27, E28, E30, E31, E32 for the left cluster) following the previous ERP studies on face processing. The following analyses were carried out for averaged ERP waveform across the five electrodes included in each cluster. The mean amplitude of N170 was measured in bilateral occipito-temporal sensor clusters within 140 to 200 ms after stimulus onset. The mean amplitude of P250 was quantified as averaged amplitude from 200 to 270 ms after stimulus onset in bilateral occipito-temporal sensor clusters. Several ERP studies on self-face recognition found the effect of self-relatedness in the long-latency positive component (LPC) [13,45]. Thus, the average amplitude of LPC was measured as averaged amplitude from 500 to 700 ms after stimulus onset in bilateral occipito-temporal sensor clusters. The mean amplitudes were entered into an ANOVA with the within-participant factors of Hemisphere (left-right) x Identity (2) × Version (3). The significance threshold was set to 0.05. Statistical analyses were carried out by R Analytic Flow (ef-prime Inc., Tokyo, Japan) and anovakun (retrieved from http://riseki.php.xdomain.jp/index.php?ANOVA%E5%90%9B, accessed on 10 July 2021). When ANOVA revealed significant interaction, its source was examined by simple main effect analysis.

Time Series Analysis by Cluster Permutation Statistics

We searched for the temporal windows within which the conditional difference is statistically significant. ERP waveforms at occipito-temporal clusters were compared between the self and other familiar faces for original, aesthetically enhanced, and degraded versions by cluster–permutation statistics [46]. In this procedure, $t$ value is computed at each time point by paired $t$ test between the ERP waveforms of the two conditions (Identity; Self vs. Familiar), and time points are determined at which the difference in ERP amplitude

between self and other familiar's faces reaches significance threshold. Then, the temporal cluster is formed by joining contiguous significant time points, and the test statistics are computed as the sum of $t$ values within the cluster. The test statistics are computed 1000 times by random permutation, i.e., randomly shuffling the conditions, through which one can obtain a null distribution of test statistics. The probability of obtaining the observed test statistics is computed based on the null distribution obtained by random permutation. Maris and Oostenveld [46] have shown that extracting significant temporal clusters by this procedure allows for sensitive detection of time windows of interest while controlling the family-wise error rate to the expected level. The family-wise error rate was adjusted to $0.05/3 = 0.0167$ because there were a total of three comparisons.

In addition to ERP waveform, the same comparisons were made for time series of global field power (GFP) [47] computed across all 61 channels by cluster–permutation statistics in the same procedure as described above to find any signs of conditional difference in the global map of scalp EEG patterns. Time series of GFP in each condition was obtained by computing across-sensor standard deviation of ERP amplitude at each time point.

Diagonal Decoding in Multivariate Analysis

Epoched EEG data were entered into multivariate analysis with the support of the Amsterdam Decoding and Modeling Toolbox [34]. The same set of epoched data from 61ch as used in the conventional ERP mean amplitude analysis was used for multivariate analysis. The procedure of multivariate analysis comprises subject-level analysis and group-level analysis. In subject-level analysis, data were first downsampled to 50 Hz (resulting in 40 time points in each trial). Linear discriminant classifier (LDC) was trained to discriminate trials of self condition from those of familiar condition at each time point. LDC was chosen because it showed superior performance in classification of ERP data compared to other classification algorithms in previous studies [33,34].

In order to avoid overfitting, cross-validation (CV) was performed by 5-fold CV procedure. In 5-fold CV, one fifth of the trials were used as test data and the remaining trials as training data. Using the training data, LDC was trained at each time point ($-100$ to 700 ms after stimulus onset) to classify self and familiar trials, and its classification performance was quantified using the test data. Every trial was used as test data once during the CV. In this step of CV, termed "diagonal decoding" [34], classification performance of LDC trained at time point $t_1$ was tested by the test data at the same time point $t_1$.

Based on the performance of test data classification, area under the curve (AUC) of the receiver operator characteristics (ROC) curve was calculated at each time point within the time widow of $-100$ to 700 ms after stimulus onset. Training of LDC was carried out separately for the three Version conditions (Original, Enhanced, and Degraded). Thus, a total of three time series of AUC was obtained for each participant in the subject-level analysis.

In the group-level analysis, time series of AUC in each Version condition was tested against chance level (AUC = 0.5) at each time point. Significant temporal cluster was searched for while controlling for the family-wise error rate by the permutation clustering approach with 1000 iterations [46]. Because a total of three comparisons were made, the significance threshold was set to $0.05/3 = 0.0167$.

Temporal Generalization Analysis

The temporal stability of the scalp EEG pattern that differentiates self and familiar condition was tested by temporal generalisation analysis [48,49]. The same set of epoched data from 61ch as used in the multivariate analysis was used for temporal generalization analysis. If the scalp EEG pattern that distinguishes self and familiar conditions is stable across time points $t_1$ and $t_2$ ($t_1$ is different from $t_2$), the model for classifying self and familiar trials trained at time point $t_1$ should succeed in classifying trials at time point $t_2$ as well. Temporal stability of neural activation that dissociates self and other familiar face processing was tested based on this logic in temporal generalization analysis.

The flow of temporal generalization analysis in the present study was essentially the same with diagonal decoding in 2.4.2.3., with one important exception. In diagonal decoding, classification performance of LDC trained at time point $t_1$ was quantified as the ability to classify test trials at the identical time point $t_1$. However, in temporal generalization analysis, performance of LDC trained at time point $t_1$ was tested at all the other temporal points as well as at $t_1$. Thus, in contrast to the case of diagonal decoding, AUC is computed for every combination of temporal points resulting in a matrix of $40 \times 40$ AUCs that is generally termed "temporal generalization matrix". In the subject-level analysis, temporal generalization matrix was computed in each of the three Version conditions. Thus, three temporal generalization matrices were obtained for each participant.

In the group-level analysis, the cluster of AUCs that is significantly different from chance level (AUC = 0.5) was searched for within the temporal generalization matrix by cluster–permutation statistics. In contrast to the cluster–permutation statistics described thus far, the cluster in the temporal generalization matrix was two-dimensional. Aside from this point, the principle underlying this procedure was the same with cluster–permutation tests for time series data [46].

## 3. Results

### 3.1. RT and Accuracy Rate

The mean and standard deviations of RT and accuracy rate in each condition are summarised in Table 1. ANOVA on RT revealed no significant effect either on the main effects or on the interaction, $Fs < 2.2$, $ps > 0.13$. There was no significant effect for accuracy rate either, $Fs < 1.45$, $ps > 0.7$.

**Table 1.** Mean and standard deviation of RT and accuracy rate in each condition. Standard deviations are in parenthesis.

| | Self | | | Other Familiar | | |
|---|---|---|---|---|---|---|
| | Original | Enhanced | Degraded | Original | Enhanced | Degraded |
| RT (ms) | 494.1 | 505.8 | 513.3 | 493.3 | 484.4 | 492.2 |
| | (73.7) | (87.0) | (104.8) | (68.9) | (84.1) | (68.3) |
| Accuracy (%) | 93.6 | 93.7 | 93.7 | 95.1 | 93.6 | 94.9 |
| | (4.9) | (4.5) | (5.9) | (4.7) | (5.4) | (5.4) |

### 3.2. ERP Results

3.2.1. ERP Mean Amplitude Analysis at Occipital Electrodes

The grand averaged waveforms in occipital sensors are shown in Figures 2 and 3. ANOVA revealed a significantly larger P100 amplitude in the right than in the left hemisphere, $F(1, 17) = 13.12$, $p = 0.002$, $\eta_p^2 = 0.43$. No other significant effects were observed for P100 amplitude, $Fs < 1.4$, $ps > 0.26$.

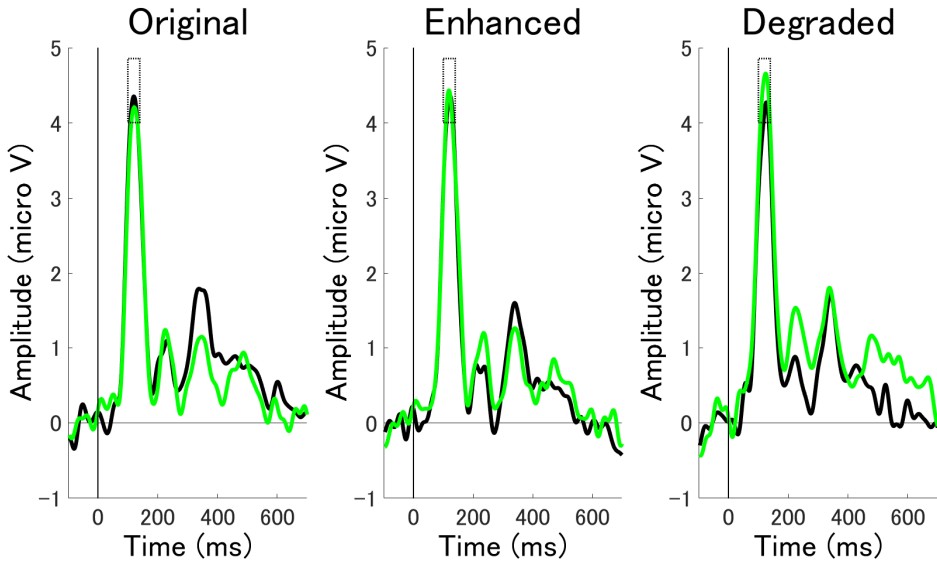

**Figure 2.** ERP waveforms recorded in each condition at left occipital sensor. The black and green lines represent ERP waveforms for self and other familiar faces, respectively. The rectangles drawn by the dotted line represent time windows used for mean amplitude measurement of P100.

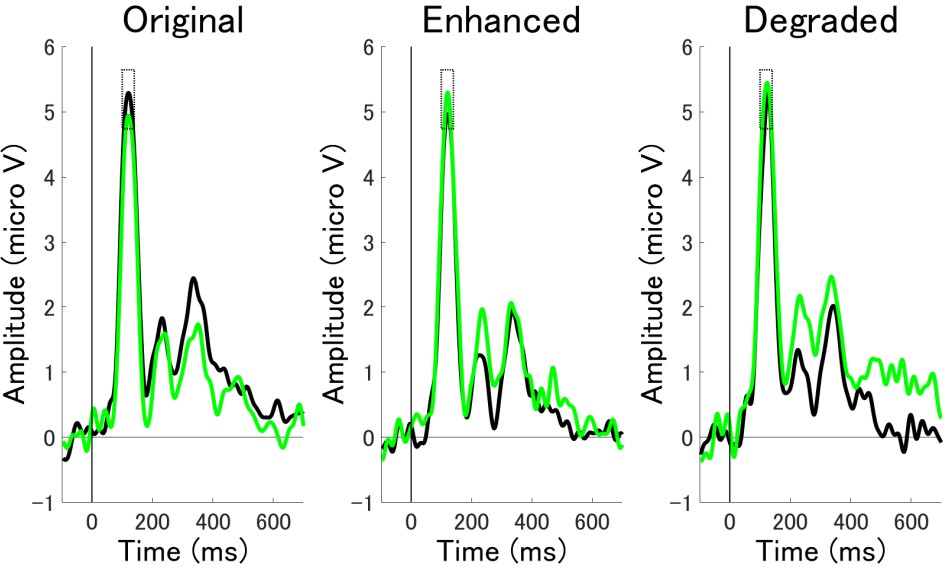

**Figure 3.** ERP waveforms recorded in each condition at right occipital sensor. The black and green lines represent ERP waveforms for self and other familiar faces, respectively. The rectangles drawn by dotted lines represent time windows used for mean amplitude measurement of P100.

### 3.2.2. ERP Mean Amplitude Analysis at Occipito-Temporal Electrode Clusters

The grand averaged waveforms in occipito-temporal electrode clusters are shown in Figures 4 and 5. ANOVA revealed a significant interaction between Identity and Version, $F(2, 34) = 4.61$, $p = 0.016$, $\eta_p^2 = 0.21$, for mean amplitude of N170. No other effects reached significance, $Fs < 2.8$, $ps > 0.07$.

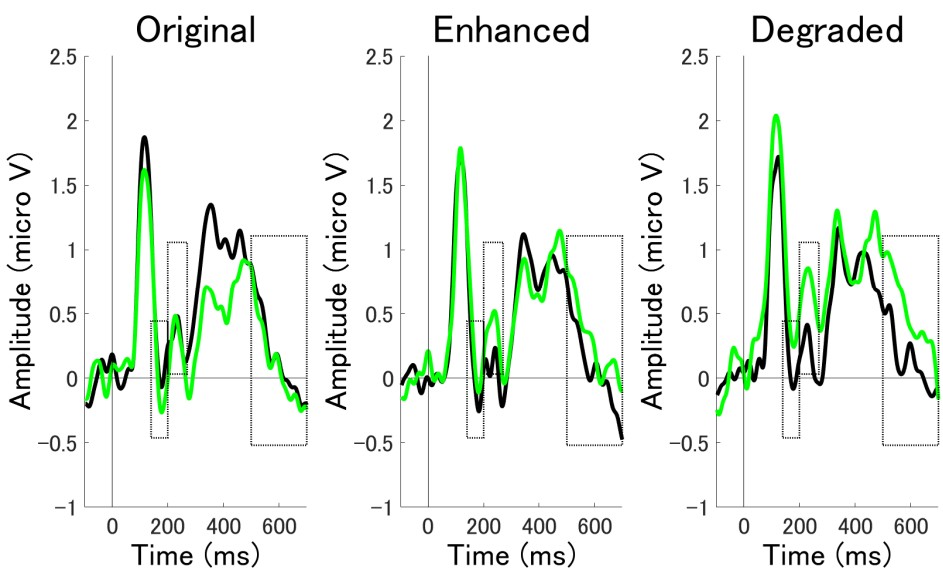

**Figure 4.** ERP waveforms recorded in each condition at left occipito-temporal electrode cluster. The black and green lines represent ERP waveforms for self and other familiar faces, respectively. The rectangles drawn by the dotted line represent time windows used for mean amplitude measurement of N170, P250 and LPC, respectively.

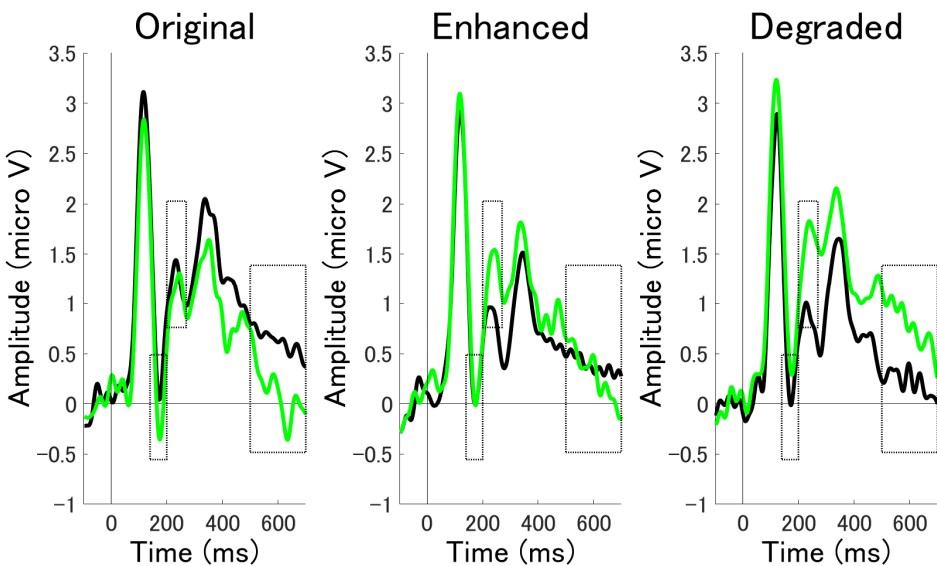

**Figure 5.** ERP waveforms recorded in each condition at right occipito-temporal electrode cluster. The black and green lines represent ERP waveforms for self and other familiar faces, respectively. The rectangles drawn by the dotted line represent time windows used for mean amplitude measurement of N170, P250 and LPC, respectively.

In order to clarify the source of the interaction, simple main effect analysis was conducted. Simple main effect analysis revealed significantly larger N170 in response to self than other familiar faces in the degraded version condition, $F(1, 17) = 7.28$, $p = 0.015$, $\eta_p^2 = 0.29$. Simple main effect of Identity did not reach significance in the other Version conditions, $Fs < 3.4$, $ps > 0.08$.

P250 amplitude was significantly larger in the right than in the left hemisphere, $F(1, 17) = 4.86$, $p = 0.041$, $\eta_p^2 = 0.22$, and significantly larger in response to other familiar than self-faces, $F(1, 17) = 7.81$, $p = 0.013$, $\eta_p^2 = 0.31$. No other effects reached significance, $Fs < 3.2$, $ps > 0.05$.

The ANOVA revealed no significant effects for LPC amplitude, $Fs < 1.9$, $ps > 0.17$.

### 3.2.3. Time Series Analysis by Cluster Permutation Statistics

The cluster permutation analysis of ERP time series revealed no statistically significant temporal clusters in the occipito-temporal electrode clusters. The waveforms of GFP are shown in Figure 6. The analysis of GFP did not reveal significant temporal clusters either.

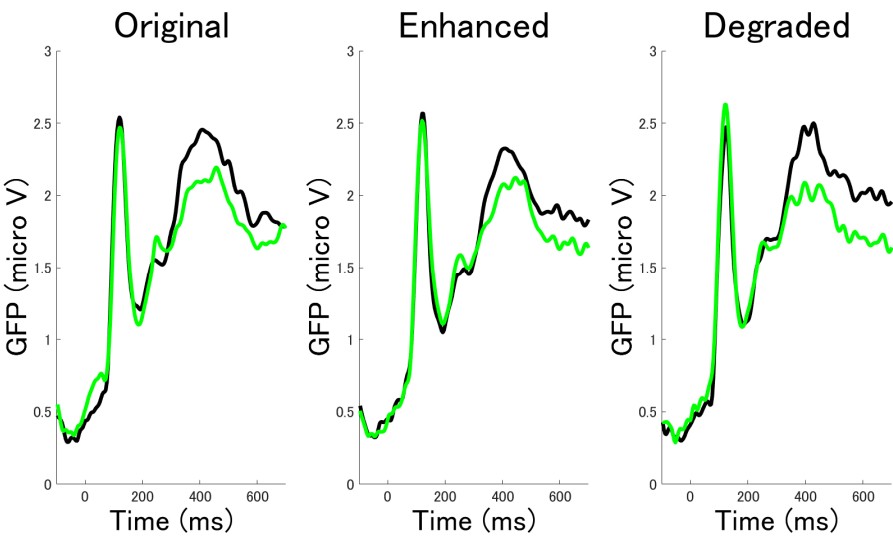

**Figure 6.** Temporal course of GFP in each condition. The black and green lines represent ERP waveforms for self and other familiar faces, respectively.

### 3.3. Diagonal Decoding Results in Multivariate Analysis

The temporal course of AUC is shown in Figure 7. As can be seen in this figure, scalp EEG patterns in self and familiar conditions can be classified from each other from 448 to 588 ms after stimulus onset for aesthetically degraded faces, *cluster p < 0.009*. There was also a significant temporal cluster for the original version of face. This temporal cluster spanned from 388 to 688 ms after stimulus onset, *cluster p < 0.001*. There was no significant or marginally significant cluster for aesthetically enhanced faces.

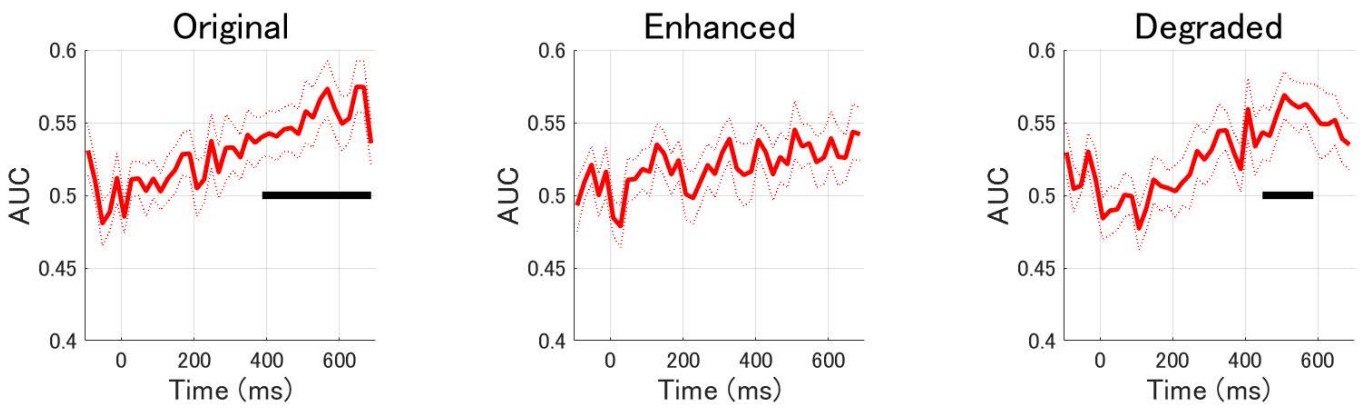

**Figure 7.** Temporal course of AUC for classification of self and other familiar faces in each condition. The black thick horizontal bar at the level of AUC = 0.5 represents the temporal window during which AUC was determined to be statistically above chance level by cluster–permutation statistics. The dotted lines represent standard errors.

### 3.4. Temporal Generalization Analysis Results

Temporal generalisation matrix and the significant cluster in each condition are described in Figure 8. A statistically significant cluster was observed at the long-latency range for original faces, *cluster p < 0.005*, but not for aesthetically manipulated versions.

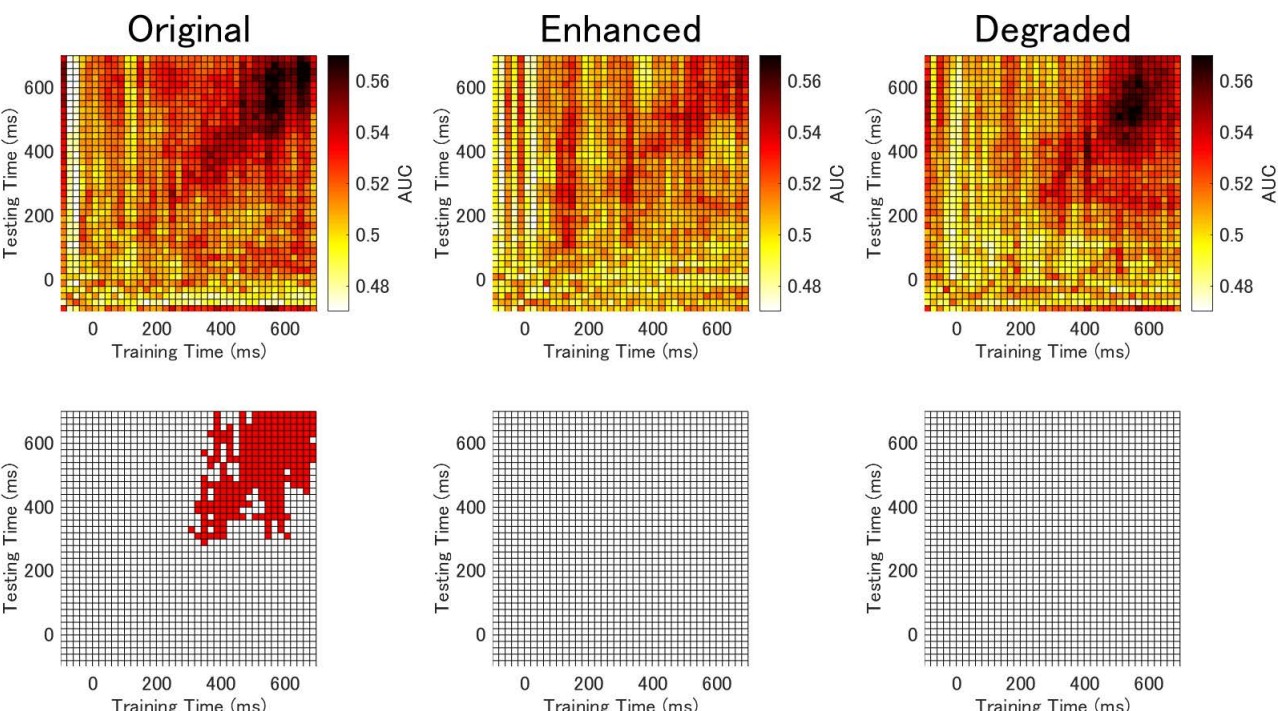

**Figure 8.** Temporal generalization matrix for classification of self and other familiar faces in each condition. In the upper panes, colour represents the values of AUC. The significant cluster is described in red in the lower panels.

*3.5. Subjective Evaluation*

The mean and standard deviations of subjective evaluations for each type of facial image are summarised in Table 2. As for attractiveness evaluation, participants rated other familiar face significantly more attractive than their own face, $F(1, 17) = 28.4$, $p < 0.001$, $\eta_p^2 = 0.63$. There was also a significant main effect of Version, $F(2, 34) = 31.2$, $p < 0.001$, $\eta_p^2 = 0.65$. Multiple comparisons revealed significantly higher attractiveness rating to enhanced than the original, $t(17) = 2.18$, *adjusted p* = 0.044, and degraded version, $t(17) = 5.99$, *adjusted p* < 0.001. Original images were rated more attractive than the degraded version, $t(17) = 6.22$, $p < 0.001$. The interaction between Identity and Version failed to reach significance, $F(2, 34) = 0.04$, $p = 0.96$, $\eta_p^2 = 0.002$.

**Table 2.** Subjective ratings given to faces in each condition. In the parenthesis are the standard deviations.

|  | **Self** | | | **Other Familiar** | | |
|---|---|---|---|---|---|---|
|  | **Original** | **Enhanced** | **Degraded** | **Original** | **Enhanced** | **Degraded** |
| Attractiveness | 9.4 | 10.5 | 5.5 | 13.2 | 14.3 | 9.1 |
|  | (3.9) | (3.3) | (3.1) | (1.5) | (2.6) | (1.9) |
| Prettiness | 8.9 | 11.1 | 5.9 | 12.4 | 14.6 | 9.3 |
|  | (3.4) | (3.7) | (3.1) | (1.4) | (2.5) | (2.1) |
| Similarity | 18.6 | 15.3 | 14.8 | 4.8 | 4.9 | 5.1 |
|  | (3.3) | (3.4) | (2.9) | (3.1) | (3.1) | (3.0) |

As for prettiness evaluation, participants rated other familiar faces as significantly prettier than their own face, $F(1, 17) = 25.6$, $p < 0.001$, $\eta_p^2 = 0.60$. There was also a significant main effect of Version, $F(2, 34) = 30.9$, $p < 0.001$, $\eta_p^2 = 0.65$. Multiple comparisons revealed significantly higher prettiness rating to enhanced than the original, $t(17) = 4.02$, *adjusted p* < 0.001, and degraded version, $t(17) = 5.91$, *adjusted p* < 0.001. Original images were rated prettier than the degraded version, $t(17) = 5.89$, *adjusted p* < 0.001. The inter-

action between Identity and Version failed to reach significance, $F(2, 34) = 0.046$, $p = 0.96$, $\eta_p^2 = 0.003$.

As for self-similarity ratings, the ANOVA revealed significant main effects of Identity, $F(1, 17) = 140.9$, $p < 0.001$, $\eta_p^2 = 0.89$, and Version, $F(2, 34) = 6.29$, $p = 0.005$, $\eta_p^2 = 0.27$. These main effects are qualified by a significant interaction between Identity and Version, $F(2, 34) = 13.57$, $p < 0.001$, $\eta_p^2 = 0.44$. A simple main effect of Version was significant in self, $F(2, 34) = 10.44$, $p < 0.001$, $\eta_p^2 = 0.38$, but not in familiar condition, $F(2, 34) = 0.42$, $p = 0.66$, $\eta_p^2 = 0.02$. Multiple comparisons revealed that the original image was rated more similar to self-face than to the enhanced, $t(17) = 3.22$, *adjusted p* $= 0.005$, and degraded version, $t(17) = 4.63$, *adjusted p* $< 0.001$. The self-similarity ratings for enhanced and degraded versions did not differ significantly between each other, $t(17) = 0.61$, *adjusted p* $= 0.55$.

## 4. Discussion and Conclusions

Self-face recognition holds a unique status in human social cognition [50,51], possibly serving as the basis of indispensable functions such as empathy [52,53]. Many people see their faces every day. However, the representation of self-face is known to be vulnerable to cognitive bias such as self-serving bias [8,53,54], which is demonstrated in several studies showing that people tend to select aesthetically enhanced versions of their face as the actual self-face [8]. The present study sought to clarify the neural basis of such a bias in self-face recognition by measuring ERPs elicited by original, aesthetically enhanced, and degraded versions of self and other familiar faces.

Conventional analysis of ERP mean amplitude revealed larger P100 in the right than in the left occipital region. Although P100 is generally considered to reflect processing of low-level visual attributes such as luminance, this component is also reported to be sensitive to face-specific information [55,56]. Taking this together with the oft-reported right-lateralization of face processing [57], presentation of facial stimuli might have enlarged P100 in the right hemisphere in the present study.

A previous study [20] examined the influence of facial familiarity on early ERP components in the occipito-temporal region and found that N170 amplitude increases and P250 amplitude decreases as facial familiarity is higher. Our observation of larger P250 amplitude in response to other familiar versus self-face replicates their findings, which probably reflects the process of matching structural information of viewed faces with stored facial representations [19,20]. In the present study, N170 amplitude was larger for self than other familiar faces, partially replicating the facial familiarity effect in [19], but puzzlingly, this trend was not observed for the original version. One plausible explanation for this null result is cultural differences. Many existing studies recruited participants from Western countries, whereas the participants of the current study were Japanese, where people accept collectivistic view [58], and familiar people are deemed as an extension of the self. Such collectivistic tendency might have blurred any differences in neural activations elicited by self and other familiar faces [59]. Another potential explanation is that the adaptation to the aesthetically manipulated version of self-face influenced processing of the original and aesthetically enhanced self-face. Recent studies using a serial dependence paradigm indicate that high-level adaptation to visual stimuli takes place after viewing an image just for a short duration [60,61]. Unfortunately, the experimental design of the present study does not afford examination of perceptual adaptation and habituation effects on ERP components because a relatively small number of trials were conducted for an ERP study. However, it is conceivable that structural encoding stages of self and other faces as reflected in N170 [19,20,62] might have been modified by the repeated exposure to aesthetically manipulated faces.

Multivariate analysis and cluster-based permutation statistics of ERP time series failed to detect statistically significant effects of facial identity in the N170 component, which indicates that a theory-driven approach of conventional analysis still has an edge over a data-driven approach in the analysis of early components. The sampling rate of the dataset was only 50 Hz, and the data from all the 61 channels were used in the multivariate

analysis. In contrast to this, data from occipto-temporal clusters sampled at 250 Hz were submitted to ERP mean amplitude analysis. One may conceive that such differences in datasets explain the failure of multivariate analysis to find any effects in early component. However, it is not the case at least in the present study. Multivariate analysis of the data in occipito-temporal clusters sampled at 250 Hz still failed to find a statistically significant effect in the N170 component as observed in ERP mean amplitude analysis.

Occipito-temporal electrodes were chosen based on the existing findings [19,20,62] to optimize the chance of finding any effects of experimental manipulation on early visual components. This is one advantage of ERP mean amplitude approach over multivariate analysis. Another reason for superior performance of ERP mean amplitude analysis is its sensitivity to short-lasting effects. A previous study [34] has pointed out the possibility that cluster-based permutation is less sensitive than alternative procedures such as FDR correction [63] in detecting small clusters of activation, which is disadvantageous particularly for the detection of ephemeral effects in the early latency range. It is of great value for the advancement of data-driven analysis of ERP data to refine a novel procedure to balance sensitivity to and controlling false-positives of meaningful but short-lasting temporal clusters.

Neither the conventional ERP analysis nor the GFP analysis revealed notable effects during the late latency range (500–700 ms after stimulus onset). At the same time, the results of diagonal decoding in multivariate analysis have shown that the pattern of scalp ERP elicited by self-face can be discriminated from that elicited by familiar faces after 500 ms for original and degraded versions. These results show the benefit of multivariate analysis in investigating electrophysiological bases of face processing.

The averaged RT for making manual responses was about 500 ms in the present task. Thus, the long-latency effect revealed by multivariate analysis is not directly linked to perceptual discrimination of facial identity, but presumably to later cognitive and evaluative stages of self-face processing. Temporal generalization analysis revealed that this effect was temporally stable for the original, but not for the degraded version of faces, which indicates that neural activation underlying the long-latency effect differs between the original and degraded versions.

As for actual self-faces, several previous studies have shown that an attentional effect of self-related information emerges around 500 ms after stimulus onset [13,19]. Thus, one explanation for the long-latency effect for original faces is the vigorous neural activations originating from the potency of self-face to capture one's visuo-spatial attention [40,41,64].

To the best of our knowledge, the present study is the first to measure ERP responses to aesthetically degraded versions of self-face. Thus, one can only make speculations about the functional significance of an identity-related effect observed for aesthetically degraded versions of faces at the long-latency range. Several behavioural studies found that self-threatening information triggers a psychological defence mechanism that protects positive self-image. Schwinghammer et al. (2006) [6] investigated how activation of negative self-conception influences evaluation of own and other's attractiveness. Their main finding was that the participants primed with negative conception avoided negative self-evaluation and gave lower attractiveness ratings to attractive others to maintain their self-esteem. Based on these findings, I tentatively propose that differential ERP pattern between self and other familiar faces in aesthetically degraded condition reflects the activation of a defensive reaction triggered by viewing aesthetically degraded self-images that are demeaning to the participants.

There are several limitations to the present study. First, only young adult females with relatively small sample size were recruited as participants, which severely limits the generalizability of the present findings. A recent study [65] reported that a larger proportion of females versus males makes an effort to edit their selfies shared on social-networking services. Thus, it is highly conceivable that males, who are less accustomed to viewing aesthetically manipulated self-representation, show neural activation different from females. Second, low-level visual features, such as global luminance and distribution

of spatial frequency power, were not strictly controlled, which leaves open the possibility that any observed effects of experimental manipulation can be explained by differences in low-level visual features, especially in cases of early ERP components. However, care was taken to ensure that effects of low-level visual features do not introduce unintentional effects into the results. First, three versions of facial images were created by warping identical skin texture. Second, self and familiar faces were presented in a folked-tier design [66]. Images of self-face served as other familiar faces for the paired participant. Thus, although potential effects of low-level features cannot be excluded entirely, I can say that such undesirable effects must have been minimized. Lastly and most importantly, findings of the present study tell little about functional significance and psychological mechanisms underlying the observed effects of aesthetic manipulation and facial identity on ERPs. For example, manipulation of facial morphology adopted in the present study probably influenced multiple aspects of facial impression other than facial beauty; large eyes and small nose in aesthetically enhanced versions are generally linked to baby schema [67,68], and a larger jaw in the degraded version is a prominent feature of facial masculinity [69,70]. However, it is unclear which dimension of facial evaluation, e.g., facial beauty, babyishness or masculinity, the observed pattern of ERP responses is linked to.

In summary, the present study found a differential pattern of neural activation in response to self and other familiar faces and that the effect of facial identity was most prominent for aesthetically degraded versions of facial images. Effects of identity were observed both in N170 and the long-latency range for the aesthetically degraded version. Importantly, the effect of self-relatedness in the long-latency range was detected only by multivariate analysis based on machine learning but not by conventional ERP amplitude or GFP analysis. In contrast, the identity effect on the N170 component was observed only by a conventional approach of ERP mean amplitude analysis. Taken together, these results suggest the necessity of integrating both theory- and data-driven approaches in a complementary manner to clarify the neural activation associated with self-face processing in its entirety.

**Funding:** This research was funded by JSPS KAKENHI Grant-in-Aid for Scientific Research (C) grant number 17K01904 and the grant from Kose Cosmetology Research Foundation to H.D. No grant no for the grant from Kose Consmetology Research Foundation.

**Institutional Review Board Statement:** The study was conducted in accordance with the Declaration of Helsinki, and approved by the ethics committee of graduate school of biomedical sciences, Nagasaki University (05053035, 12/5/2006).

**Informed Consent Statement:** Informed consent was obtained from all participants involved in the study.

**Data Availability Statement:** The data presented in this study are available on reasonable request from the corresponding author.

**Acknowledgments:** Interim analysis of a preliminary experiment of this research had formerly been published in the Annual Report of Cosmetology (2012).

**Conflicts of Interest:** The authors declare no conflict of interest.

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
