# Peer review of "Multivariate ERP Analysis of Neural Activations Underlying Processing of Aesthetically Manipulated Self-Face"

_applsci, doi:10.3390/app122413007_

Round 1
Reviewer 1 Report
The research topic is very interesting, due to several reasons: (1) it covers the topic of facial representation in electrophysiology research which is an interesting phenomenon itself, (2) deals with the self- vs. others- perspective - it is very important to understand how this cognitive process works, in many applications, from general basic understanding of the cognitive process, through body ownership, to applications in neuroengineering and neurorobotics, (3) studies machine learning application in ERP research, but not for online EEG classification what is mainly used in BCI applications, but instead to increase the statistical power in ERP cognitive research, which gains more and more popularity in the field.
Please find below my comments and suggestions which, I hope, will help improve the research.
1.
In the introduction the Author refers to the peak amplitude and peak latency as the method used in ERP research. However it’s worth noting that, even though these methods are still used in many research, there have been better approaches used for many years now, like mean or fractional amplitude/latency or permutation cluster-based tests, which doesn’t require a priori specification of ERP components of interest (e.g. Luck, 2014, Maris and Oostenveld, 2007).
Moreover, I think it would be beneficial to underline and provide more references for the studies using machine learning approach in EEG, but not for classification or regression of the ongoing EEG signal online, what has been used for decades mainly for BCI applications in clinical studies (e.g. for classify pathological features of the EEG signal), but for the ERP analysis in cognition in order to improve the statistical power.
2.2.
Were the images controlled for possible differences in physical image parameters between conditions (mean intensity, contrast, etc)? It is very important in ERP research, especially for the early potentials.
2.3.1
Please provide information about the model and manufacturer of EEG system (amplifier and EEG cap) and presentation software. I believe Netstation software was used for data acquisition. Please add more information. How many channels were used? What was the ground and the reference? Please describe the preparation phase.
It would be clearer if this subsection contained only information about EEG measurement and the information about experimental design was described in another subsection. In this subsection more data about the design should be presented, i.e.:
- was the fixation cross present also in the face images? Was there an equal number of trials per each condition in each block?
- Since (if I understand correctly) there were only 6 separate images presented to each participant, how was the habituation controlled for (or verified)?
- How was the pseudorandomization performed (i.e. please explain why not randomization)?
- What was the time of each block, the break and the experiment in total?
- Please provide more information about participants’ responses – how were the identity conditions discriminated (i.e. left/right finger, or different fingers of the same hand)? Were these counterbalanced to avoid differences between conditions due to different finger/hand movements (i.e. across blocks or across participants)?
- Were the participants instructed to discriminate the faces by the button press as soon as possible after 1 s following stimulus onset or as soon as possible after the stimulus onset? If it was the 2nd – then there might be a confounding factor due to finger movement (especially if the movements were not counterbalanced).
- The Author states: “They were told to classify aesthetically enhanced and degraded version of self-face and the original version as their own face.” Please provide more information about this second judgement test. When was is performed? How did the participants respond? Etc.
- Regarding the sentence above – it is not clear to me whether there were only 2 classes: aesthetically changed (collapsed across enhanced and degraded) and original, or three classes
- Please provide full ERP measurement design (usually it’s best to show it in a figure presenting an exemplary trial(s)), i.e.: fixation (500 ms) -> image (1s) -> response…etc
2.3.2
What was the purpose of creating two distinct behavioral measures to evaluate the attractiveness: (not)attractive and (not)pretty? Is there a psychological/cognitive difference in the definition of these two descriptions? If so, please provide the rationale.
2.4.1
Please provide the software used for the statistical analyses. Please describe the ANOVA design in this section.
2.4.2
Please provide which software was used for EEG pre-processing.
Please provide the details of the ICA correction. Which type of artifacts were corrected? How many components were removed on average across participants? Which were the criteria for identifying the components as artifacts? Was the exclusion performed based on manual visual inspection of the components or was there any kind of (semi)automatic tool used? Was the rank deficiency due to average reference fixed?
Please provide the average number of trials per each condition across the participants after pre-processing.
2.4.2.1
Which electrodes fell within the electrode cluster? What was the selection criterion?
The author used two different methods to measure amplitude for different waveforms: peak amplitude for P100 and N170 and mean amplitude for P220 and LPC. There should be only one technique for all the components chosen and the mean amplitude is preferred (see e.g. Luck, 2014 for reference).
What was the alpha level for the traditional ERP analyses and what was the MCP correction method used?
2.4.2.2
Please provide more information on the permutation cluster analysis, e.g. time window and electrode clusters, method to calculate the significance probability, correction method, test statistic for the evaluation under the permutation distribution, method and minimal number of channels to determine the neighbours, number of permutations, etc. + used method/tool/software (and references, if applicable)
2.4.2.3.
I’m not sure I fully follow the multivariate analysis combined with the cluster permutation approach. If I understand correctly – there was a machine learning method used to classify the conditions, and then the cluster permutation approach here was used to evaluate the significance of the obtained results?
In general, please broaden the description of the machine learning approach – e.g. what type of algorithm was used, what were the parameters, how were the training and testing sets assigned, etc.
3.1.
Which effect do these statistical results apply to (since there were two main effects and one interaction)?
Figure 2.
Please correct the legend (enhanced is twice instead of degraded). Also the legend takes much space, while x and y titles are hard to read. Maybe it would better to rearrange the figure. Also please consider adding some kind of information of the peaks and time windows of interest (by indicating by an arrow the analyzed peaks or a rectangle in the background showing the time window of interest – depending which measure the Author decides to use)
3.3. and Figure 4
Bottom panel – why are these uncorrected p-values?
3.3
Please explain in more details the temporal generalisation matrices – what do the values of the pixels mean and how to interpret them.
Figure 5
The labels are very small
The scale of the color bar for the ‘enhanced’ condition is different than the other two
Discussion
The sample size of this research is very poor (only 9 participants), which should be discussed as a limitation and a possible reason for not observing significant effects in the traditional and permutation-based ERP analyses.
Please discuss the possible neural estimate of the significant effect in the late time window (590-680 ms).
The sections lacks the discussion of the lateralisation effect of P100 (which is odd), observed only for the traditional ERP analysis.
Author Response
Response to Reviewer #1
Below I summarized my responses the comments by reviewer #1. The corrected sentences are highlighted in red in the manuscript.
Please note that the overall results have changed to some extent (especially those of conventional ERP analysis), after I re-analyzed the data by fixing the problem of rank deficiency in ICA and adopting mean amplitude approach instead of peak amplitude/latency analysis following your comments. Consequently, I ended up renovating Results and Discussion section. I sincerely apologize for the confusion.
Comment 1:
However it’s worth noting that, even though these methods are still used in many research, there have been better approaches used for many years now, like mean or fractional amplitude/latency or permutation cluster-based tests, which doesn’t require a priori specification of ERP components of interest (e.g. Luck, 2014, Maris and Oostenveld, 2007).
Moreover, I think it would be beneficial to underline and provide more references for the studies using machine learning approach in EEG, but not for classification or regression of the ongoing EEG signal online, what has been used for decades mainly for BCI applications in clinical studies (e.g. for classify pathological features of the EEG signal), but for the ERP analysis in cognition in order to improve the statistical power.
Response: Thank you for the suggestions. Following your comments, I mentioned existing methods of ERP analysis without a priori specification f components of interest and machine learning approach of ERP/MEG analyses besides BCI application.
“In addition to the conventional analysis, various methods for ERP and EEG analysis had been developed; fractional latency/amplitude measurement [21] aims to increase robustness of ERP measurement against fluctuating noise, and application of principal component analysis to ERP time-series [22, 23] to disentangle overlapping components. Another type of approaches tries to detect ERP responses sensitive to experimental manipulation without a priori specification of ERP components and latency range of interest. These include ERP spatio-temporal cluster-based permutation of scalp ERP field [24] and microstate segmentation analysis [25]. Recent surge of interest in machine learning led to utilization of machine learning algorithms in ERP/EEG analysis. This line of research mainly focused on prediction of internal states [26, 27, 28, 29]. Besides practical application, machine learning approach had also given novel insights into neurophysiological mechanism of perceptual and cognitive processing [30, 31]. ”
Comment 2:
Were the images controlled for possible differences in physical image parameters between conditions (mean intensity, contrast, etc)? It is very important in ERP research, especially for the early potentials.
Response:
They were not explicitly controlled for such visual attributes as frequency distributions, contrast, and luminance. Because three versions of facial stimuli (original aesthetically degraded, aesthetically enhanced) were created by warping, and self and familiar faced are presented in folked design, I think that any effects of low-order visual characteristics were minimal. Still, as you pointed out, it is a concern for interpretation of ERP effects especially in early visual components. Thus, I mentioned this point as one limitation of this study.
“Second, low-level visual features, such as global luminance and distribution of spatial frequency power, were not strictly controlled, which leaves open the possibility that any observed effects of experimental manipulation can be explained by differences in low-level visual features especially in cases of early ERP components. However, care was taken to ensure that effects of low-level visual features do not introduce unintentional effects into the results. First, three versions of facial images were created by warping identical skin texture. Second, self and familiar faces were presented in a folked-tier design [65]; Images of self faces served as familiar other’s face for the paired participant. Thus, though potential effects of low-level features cannot be excluded entirely, I can say that such undesirable effects must have been minimized.”
Comment 3:
Please provide information about the model and manufacturer of EEG system (amplifier and EEG cap) and presentation software. I believe Netstation software was used for data acquisition. Please add more information. How many channels were used? What was the ground and the reference? Please describe the preparation phase.
Response:
I described these pieces of information accordingly.
“We wrote our experiments in Matlab, using the Psychophysics Toolbox extensions [42].”
“EEG referenced to Cz was recorded by Geodesic 64ch EEG System (NetAmps 300, Electrical Geodesics Inc, Eugene, Oregon, USA)”
“The data from two electrodes (E62, E63) on the facial surface mainly for detection of eye-movement artifacts and those near the tragus (E23, E55) were deleted from the dataset, and the data from the remaining 61 channels were entered into further analysis.”
Comment 4:
It would be clearer if this subsection contained only information about EEG measurement and the information about experimental design was described in another subsection.
Response:
The subsections were restructured according to your recommendation.
Comment 5:
In this subsection more data about the design should be presented, i.e.:
- was the fixation cross present also in the face images? Was there an equal number of trials per each condition in each block?
- How was the pseudorandomization performed (i.e. please explain why not randomization)?
- What was the time of each block, the break and the experiment in total?
- Please provide more information about participants’ responses – how were the identity conditions discriminated (i.e. left/right finger, or different fingers of the same hand)? Were these counterbalanced to avoid differences between conditions due to different finger/hand movements (i.e. across blocks or across participants)?
- Were the participants instructed to discriminate the faces by the button press as soon as possible after 1 s following stimulus onset or as soon as possible after the stimulus onset? If it was the 2nd – then there might be a confounding factor due to finger movement (especially if the movements were not counterbalanced).
- Please provide full ERP measurement design (usually it’s best to show it in a figure presenting an exemplary trial(s)), i.e.: fixation (500 ms) -> image (1s) -> response…etc
Response: Details about these points are described in the revised manuscript.
“After the disappearance of the fixation cross, a stimulus face subtending about 7.8 deg in width and 7.8 in height was presented for 1 s.”
“Pseudo-randomization of trials was achieved by the following procedure. First, trials of the six conditions were randomized to form a sub-cluster of six trials. Each block is created by concatenating 15 sub-clusters of trials. Thus, the number of trials in each of the six conditions was equated to be fifteen within each block. Complete randomization of 540 trials was not adopted to avoid the situation that trials of identical trials are repeated many times in succession.”
“Each block lasted for about 2.3 mins. The length of brief rest varied across participants, but was generally shorter than 1.5 mins. The rest period after the third block lasted for about 5 mins for checking and, if necessary, lowering contact impedance.”
“Participants made their responses by key-pressing by left or right thumb. The correspondence between key and response (self or other’s face) was counterbalanced across pairs. If they did not answer within the period of facial stimulus presentation, response in the trial was treated as incorrect.”
The temporal sequence of stimulus presentation is now described in Figure 1.
Comment 6:
- Since (if I understand correctly) there were only 6 separate images presented to each participant, how was the habituation controlled for (or verified)?
Response: Thank you for raising an important point. As in many ERP studies, stimuli in the identical condition had been repeated many times (in the current study 90 times) throughout the experiment, which surely have made participants habituated or perceptually adapted to the stimuli. Unfortunately, the current study was not designed to test habituation/ adaptation effect itself and contained too few trials to examine this point. Thus, I mentioned this as one limitation of the present study.
“Unfortunately, the experimental design of the present study does not afford examination of perceptual adaptation and habituation effect on ERP components because relatively small number of trials were conducted for an ERP study. However, it is conceivable that structural encoding stages of self and other’s face as reflected in N170 [19, 20, 62] processing might have been modified by the repeated presentation of aesthetically manipulated faces.”
Comment 7:
- The Author states: “They were told to classify aesthetically enhanced and degraded version of self-face and the original version as their own face.” Please provide more information about this second judgement test. When was is performed? How did the participants respond? Etc.
- Regarding the sentence above – it is not clear to me whether there were only 2 classes: aesthetically changed (collapsed across enhanced and degraded) and original, or three classes
Response:
Details about these points are described in the revised manuscript. As for the second point you raised, facial stimuli of all the three versions (Original, Enhanced, Degraded) were presented in the second experiment as well.
“The participants evaluated the presented face by moving the trackbars to the location closest to their subjective evaluation. After moving the trackbars to the preferred location, participants clicked “proceed” button just below the trackbars. Clicking the button started the next trial. Each of the Identity (2) x Version (3) = six types of faces were evaluated twice, resulting in 12 experimental trials. The order of stimulus presentation was pseudo-randomly determined.”
Comment 8:
2.3.2
What was the purpose of creating two distinct behavioral measures to evaluate the attractiveness: (not)attractive and (not)pretty? Is there a psychological/cognitive difference in the definition of these two descriptions? If so, please provide the rationale.
Response:
In Japanese language, many aspects of favorableness are expressed by the adjective “pretty” (Kawaii). Thus, I thought that taking measure of “prettiness” is useful in measuring subjective evaluation not captured by “beauty” measurement. I stated this rationale in the manuscript.
““Pretty-Not Pretty” evaluation was obtained because “Kawaii”, the Japanese word for “Pretty”, is used to express many aspects of favorable impression [43]. Thus, measurement of “Pretty-Not Pretty” evaluation is expected to reveal aesthetic evaluation not captured by “Attractive-Unattractive” dimension.”
Comment 9:
2.4.1
Please provide the software used for the statistical analyses. Please describe the ANOVA design in this section.
2.4.2
Please provide which software was used for EEG pre-processing.
Please provide the average number of trials per each condition across the participants after pre-processing.
2.4.2.1
Which electrodes fell within the electrode cluster? What was the selection criterion?
Response: Details for these points are now described in the revised manuscript accordingly.
“an analysis of variance (ANOVA) with within-participant factors of Identity (2) and Version (3).”
“Statistical analyses were carried out by R Analytic Flow (ef-prime Inc, Tokyo, Japan) and anovakun (retrieved from http://riseki.php.xdomain.jp/index.php?ANOVA%E5%90%9B). Significance threshold of multiple comparison was adjusted by Modified Sequentially Rejective Bonferroni procedure.”
“The significance threshold was set to 0.05.”
“EEG data was pre-processed using EEGLab [44].”
“bilateral occipital sensors (E39 for the right and E35 for the left) within 100 to 140 ms”
“The left and right occipito-temporal clusters each included five electrode locations (E40, E42, E45, E44, E43 for the right cluster, and E27, E28, E30, E31, E32 for the left cluster) following the previous ERP studies on face processing.”
“After pre-processing, 69.7 trials on average were retained for further analyses; In self condition, 69.4, 71.0, 70.8 trials for original, enhanced and degraded versions, whereas in familiar condition, 67.8, 69.1, and 70.3 trials for original, enhanced and degraded versions respectively.”
Comment 10:
Please provide the details of the ICA correction. Which type of artifacts were corrected? How many components were removed on average across participants? Which were the criteria for identifying the components as artifacts? Was the exclusion performed based on manual visual inspection of the components or was there any kind of (semi)automatic tool used? Was the rank deficiency due to average reference fixed?
Response: I described more details on the methodology of IC removal accordingly.
“After decomposition, ICs were checked by visual inspection and those judged to reflect blink and motion artifact based on scalp distribution and temporal fluctuation were removed from the data, resulting in removal of on average 3.9 ICs.”
While checking rank deficiency, I found that mistake in the order of applying channel deletion (deletion of four channels) led to rank deficiency. Consequently, ICA generated apparently abnormal results in some datasets. This problem was fixed by the following manner. First, EEG was average-referenced and four channels outside the focus of interest were deleted (This way, the problem of rank deficiency is avoided). Thereafter, ICA was carried out.
“(each time-point covering 4 ms), bandpass filtered (0.1-30Hz) and re-referenced to average reference. The data from two electrodes (E62, E63) on the facial surface mainly for detection of eye-movement artifacts and those near the tragus (E23, E55) were deleted from the dataset, and the data from the remaining 61 channels were entered into further analysis.”
Fixing rank deficiency changed overall results (especially those of ERP peak analysis). Thus, I fully revised the Results section and Discussion accordingly.
Comment 11:
There should be only one technique for all the components chosen and the mean amplitude is preferred (see e.g. Luck, 2014 for reference).
What was the alpha level for the traditional ERP analyses and what was the MCP correction method used?
Response:
I re-analyzed the amplitude by mean amplitude approach following your recommendation, The latency was also re-computed by mor fractional latency method. I described detailed method of post hoc analysis including the procedure of threshold correction in multiple comparisons.
“Significance threshold of multiple comparison was adjusted by Modified Sequentially Rejective Bonferroni procedure.”
“When ANOVA revealed significant interaction, its source was examined by simple main effect analysis.”
“The significance threshold was set to 0.05.”
Comment 12:
2.4.2.2
Please provide more information on the permutation cluster analysis, e.g. time window and electrode clusters, method to calculate the significance probability, correction method, test statistic for the evaluation under the permutation distribution, method and minimal number of channels to determine the neighbours, number of permutations, etc.
Response: I described more details about the methodology of cluster permutation accordingly. Spatial clustering across channels was not conducted in the present study. Thus, “minimal number of channels to determine the neighbours” is not specified in the text. In clustering of temporal generalization matrix, contiguous (adjascent) significant time-points were jointed to form cluster.
“In this procedure, t-value is computed at each time-point by paired t-test between the ERP waveforms of the two conditions (Identity; Self vs Familiar), and time-points are determined at which the difference in ERP amplitude between self and familiar other’s faces reaches significance threshold. Then, temporal cluster is formed by joining contiguous significant time-points, and the test statistics is computed as the sum of t-values within the cluster. The test-statistics is computed for 1,000 times by random permutation, i.e. randomly shuffling the conditions, through which one can obtain null distribution of test-statistics. The probability of obtaining the observed test-statistics is computed based on the null distribution obtained by random permutation. Maris and Oostenveld [46] have shown that extracting significant temporal clusters by this procedure allows sensitive detection of time-windows of interest while controlling the family-wise error rate to expected level.”
Comment 12:
2.4.2.3.
I’m not sure I fully follow the multivariate analysis combined with the cluster permutation approach. If I understand correctly – there was a machine learning method used to classify the conditions, and then the cluster permutation approach here was used to evaluate the significance of the obtained results?
In general, please broaden the description of the machine learning approach – e.g. what type of algorithm was used, what were the parameters, how were the training and testing sets assigned, etc.
Response: I described more details about the methodology of machine learning adopted in the present study including the procedure of cross validation.
“The procedure of multivariate analysis is comprised of subject-level analysis and group-level analysis. In subject-level analysis, data was first down-sampled to 50 Hz (resulting in 40 time points in each trial). Linear discriminant classifier (LDC) was trained to discriminate trials of self condition from those of familiar condition at each time point. LDC was chosen because it showed superior performance in classification of ERP data compared to other classification algorithms in previous studies [33, 34].
In order to avoid overfitting, cross-validation (CV) was performed by 5-fold CV procedure. In 5-fold CV, one fifth of the trials were used as test data and the remaining trials as training data. Using the training data, LDC was trained at each time point (-100 to 700 ms after stimulus onset) to classify self and familiar trials, and its classification performance was quantified using the test data. Every trial was used as test data once during the CV. In this step of CV termed ‘diagonal decoding’ [34], classification performance of LDC at time point t1 was tested by the test data at the same time-point t1.
Based on the performance of test data classification, area under curve (AUC) of receiver-operator characteristics (ROC) curve was calculated at each time point within the time widow of -100 to 700 ms after stimulus onset. Training of LDC was carried out separately for the three Version conditions (Original, Enhanced, and Degraded). Thus, in total of three time-series of AUC was obtained for each participant in the subject-level analysis.
In the group-level analysis, time-series of AUC in each Version condition was tested against chance level (AUC = 0.5) at each time point. Significant temporal cluster was searched for while controlling the family-wise error rate by the permutation clustering approach with 1,000 iterations [46].”
Comment 13:
3.1.
Which effect do these statistical results apply to (since there were two main effects and one interaction)?
Response:
“either in the main effects or the interaction”
Comment 14:
Figure 2.
Please correct the legend (enhanced is twice instead of degraded). Also the legend takes much space, while x and y titles are hard to read. Maybe it would better to rearrange the figure. Also please consider adding some kind of information of the peaks and time windows of interest (by indicating by an arrow the analyzed peaks or a rectangle in the background showing the time window of interest – depending which measure the Author decides to use)
Figure 5
The labels are very small
The scale of the color bar for the ‘enhanced’ condition is different than the other two
Response:
Thank you for your suggestions. I corrected the figures according to your comments. Please see Figure 2 to 5.
Comment 15:
3.3. and Figure 4
Bottom panel – why are these uncorrected p-values?
Response:
After careful consideration, I reached the conclusion that presentation of uncorrected p-values are of limited value for readers. Thus, I deleted the lower panels entirely.
Comment 16:
3.3
Please explain in more details the temporal generalisation matrices – what do the values of the pixels mean and how to interpret them.
Response:
I described more details about the concept interpretation and methodology of temporal generalization matrix.
“If scalp EEG pattern that distinguishes self and familiar conditions is stable across time points t1 and t2 (t1 is different from t2), the model for classifying self and familiar trials trained at time-point t1 should succeed in classifying trials at time point t2 as well. Temporal stability of neural activation that dissociates self and familiar other’s face processing was tested based on this logic in temporal generalization analysis.
The flow of temporal generalization analysis in the present study was essentially the same with diagonal decoding in 2.4.2.3. with one important exception. In diagonal decoding, classification performance of LDC trained at time point t1 was quantified as the ability to classify test trials at the identical time point t1. But in temporal generalization test, performance of LDC trained at time point t1 was tested at all the other temporal points as well as t1. Thus, in contrast to the case of diagonal decoding, AUC is computed for every combination of temporal points resulting in a matrix of 40 x 40 AUCs that is generally termed ‘temporal generalization matrix’. In the subject-level analysis, temporal generalization matrix was computed in each of the three Version conditions. Thus, three temporal generalization matrices were obtained for each participant.
In the group-level analysis, cluster of AUCs that is significantly different from chance level (AUC = 0.5) was searched for within temporal generalization matrix by cluster-permutation statistics. In contrast to the cluster-permutation statistics described thus far, the cluster in temporal generalization matrix is two-dimensional. Aside from this point, the principle underlying this procedure and the test-statistics are the same with cluster-permutation tests for time-series data [46]. ”
Comment 17:
Discussion
The sample size of this research is very poor (only 9 participants), which should be discussed as a limitation and a possible reason for not observing significant effects in the traditional and permutation-based ERP analyses.
Response:
Actually, the number of participants whose data are used in the analysis was eighteen (2 participants per each pair x 9 pairs), I made this point clearer in the manuscript.
“in total of 18 participants;”
Comment 18:
Please discuss the possible neural estimate of the significant effect in the late time window (590-680 ms).
The sections lacks the discussion of the lateralisation effect of P100 (which is odd), observed only for the traditional ERP analysis.
Response:
Thank you for raising important points. I discussed these points based on the existing literature. Without further analysis like source localization, I found it too speculative to discuss neural sources of long-range effect. Thus, I refrained from discussing this point.
“Temporal generalization analysis revealed that this effect was temporally stable for the original, but not for the degraded version of faces, which indicates that neural activation underlying the long-latency effect differs between original and degraded versions. ”
“Though P100 is generally considered to reflect processing of low-level visual attributes such as luminance, this component is also reported to be sensitive to face-specific information [55, 56]. Taking this together with the oft-reported right-lateralization of face processing [57], presentation of facial stimuli might have enlarged P100 in the right hemisphere in the present study.”

Reviewer 2 Report
The study examined self versus other familiar face effects across three conditions, including original, aesthetically enhanced, and aesthetically degraded. The author did not find any differential effects using conventional ERP and univariate time-series analysis between self vs other familiar face in the occipital and occipito-temporal electrodes. Multivariate machine learning methods using linear discriminant classifier and temporal generalization analysis showed differential effects only in the degraded condition. The author concluded that machine learning methods offer a useful approach in detecting neural mechanism underlying self face processing. Although I think the results of this study is not very generalizable due to the experimental design, I can see the value of the study as a demonstration to examine ERP effects that may not be evident based on conventional methods. Most of my questions are related to the methods. I also have a few concerns about the interpretation of their main findings.
Introduction:
-It remains unclear to me why others' familiar faces were chosen, rather than strangers' faces. There is potentially an interesting and important point of this design, but this was not fully elaborated in the introduction
Methods:
-in 2.1 Should state explicitly that the total number of subjects is N = 18, if I understood it correctly. Also why only females?
-in 2.3.1 How many channels does the EEG have? Any eye electrodes or reference electrodes and where are they located?
-in 2.3.1 How did they provide their answers? by two buttons under right hand? please clarify
-in 2.3.2 It seems that "attractiveness" and "prettiness" were quite similar metrics. What was the reason to choose these matrices?
-in 2.4 If the data was downsampled to 250 Hz, how many time points were actually entered into the analysis? And how much time does each time point cover (supposedly 4 ms since it was downsampled)? especially for 2.3.2.2. and 2.4.2.3 to evaluate the stability of the model and how many comparisons were performed.
-in 2.4.2 What type of processing was used to remove artifacts using ICA? "By performing ICA" does not remove artifacts (and what kind of artifacts).
-in 2.4.2.1 What is P220? In the introduction only P250 was mentioned.
-in 2.4.2.1 What specific electrodes (that constituted "sensor clusters") were included for analysis?
-in 2.4.2.2 Did GFP include all scalp electrodes (how many?), or just the occipital and occipito-temporal electrodes?
-in both 2.4.2.2 and 2.4.2.3 What kind of statistics (F tests, t tests, etc.) for the cluster permutation statistics? Please be specific about "conditional difference" since this could be between Versions or Identities. How many electrodes did you include in the analysis versus if you only included those averaged "sensor clusters", which needs to be clarified? More clarification is needed to describe the methods to avoid any misunderstanding, such as how many time points, how many electrodes, etc.
- in 2.4.2.3 what kind of algorithm was used to control for multiple comparisons? Since you ran tests over multiple time points this needs to be clarified for cluster-based analysis. What is condition A vs condition B, Identity or Version? Also, It is unclear to me why these particular machine learning methods were chosen given that there are so many different methods available. Please provide ratione to help readers understand better.
Results:
-Several figures have very tiny fonts and are unintelligible in print (Figure 2, Figure 5 5.). Please adjust.
Conclusions:
-The self vs other in degraded condition shows this later effect around 600ms post stimulus onset, which seems to occur after response (around 492-513 in average). Therefore it seems that this effect does not really entail necessary neural activity for self vs other face processing? How would you explain that?
-could aesthetically degrade versions be explained by more masculine features? Then the results would suggest processing between masculine (degraded) versus feminine (enhanced) features of self vs other's faces.
-Only females were included and there is no clear rationale. The results can hardly be generalizable to those who are not young females.
Author Response
Response to Reviewer #2
Below I summarized my responses the comments by reviewer #2. The corrected sentences are highlighted in red in the manuscript.
Please note that the overall results have changed to some extent (especially those of conventional ERP analysis), after I re-analyzed the data by fixing the problem of rank deficiency in ICA and adopting mean amplitude approach instead of peak amplitude/latency analysis following the comment by Reviewer #1. Consequently, I ended up in renovating Results and Discussion section. I sincerely apologize for the confusion.
Comment 1:
Introduction:
-It remains unclear to me why others' familiar faces were chosen, rather than strangers' faces. There is potentially an interesting and important point of this design, but this was not fully elaborated in the introduction
Response:
Self and stranger’s face differ greatly in the degree of perceptual familiarity, which makes it hard to decide whether any effects obtain in comparison between self vs unfamiliar faces are attributable to self-relatedness or familiarity. I used familiar other’s face instead of total stranger’s face to mitigate this problem. I made this rationale clearer in the Introduction.
“Familiar other’s face was used instead of complete stranger’s face to mitigate the influences of perceptual familiarity on electrophysiological responses.”
Comment 2:
Methods:
-in 2.1 Should state explicitly that the total number of subjects is N = 18, if I understood it correctly. Also why only females?
Response:
The data had been collected about 10 years ago. Back then, Japanese females were generally more keenly self-conscious about their appearances than males (things have changed abit since then). This is why I focused on females’ responses. I made this point clearer in the revised manuscript.
“in total of 18 participants;”
“Only females were recruited as participants because previous studies have pointed out that females tend to be self-conscious of their physical appearance [16, 17].”
Comment 3:
-in 2.3.1 How many channels does the EEG have? Any eye electrodes or reference electrodes and where are they located?
-in 2.3.1 How did they provide their answers? by two buttons under right hand? please clarify
into the analysis? And how much time does each time point cover (supposedly 4 ms since it was downsampled)? especially for 2.3.2.2. and 2.4.2.3 to evaluate the stability of the model and how many comparisons were performed.
-in 2.4 If the data was downsampled to 250 Hz, how many time points were actually entered
Response:
More details are described on these points in the revised version.
“EEG referenced to Cz was recorded by Geodesic 64ch EEG System (NetAmps 300, Electrical Geodesics Inc, Eugene, Oregon, USA)”
“Participants made their responses by key-pressing by left or right thumb. The correspondence between key and response (self or other’s face) was counterbalanced across pairs. If they did not answer within the period of facial stimulus presentation, response in the trial was treated as incorrect.”
“Because the data was down-sampled to 250Hz, each epoch contained 200 time points.”
“In subject-level analysis, data was first down-sampled to 50 Hz (resulting in 40 time points in each trial).”
“Thus, in contrast to the case of diagonal decoding, AUC is computed for every combination of temporal points resulting in a matrix of 40 x 40 AUCs that is generally termed ‘temporal generalization matrix’.”
Comment 4:
-in 2.3.2 It seems that "attractiveness" and "prettiness" were quite similar metrics. What was the reason to choose these matrices?
Response:
In Japanese language, many aspects of favorableness is expressed by the adjective “pretty” (Kawaii). Thus, I thought that taking measure of “prettiness” is useful in measuring subjective evaluation not captured by “beauty” measurement. I stated this rationale in the manuscript.
““Pretty-Not Pretty” evaluation was obtained because “Kawaii”, the Japanese word for “Pretty”, is used to express many aspects of favorable impression [43]. Thus, measurement of “Pretty-Not Pretty” evaluation is expected to reveal aesthetic evaluation not captured by “Attractive-Unattractive” dimension.”
Comment 5:
-in 2.4.2 What type of processing was used to remove artifacts using ICA? "By performing ICA" does not remove artifacts (and what kind of artifacts).
Response: I described more details on the methodology of IC removal accordingly.
“After decomposition, ICs were checked by visual inspection and those judged to reflect blink and motion artifact based on scalp distribution and temporal fluctuation were removed from the data, resulting in removal of on average 3.9 ICs.”
Comment 6:
-in 2.4.2.1 What is P220? In the introduction only P250 was mentioned.
Response: I decided to stick to the term “P250” throughout the manuscript to avoid confusion.
Comment 7:
-in 2.4.2.1 What specific electrodes (that constituted "sensor clusters") were included for analysis?
-in 2.4.2.2 Did GFP include all scalp electrodes (how many?), or just the occipital and occipito-temporal electrodes?
-in both 2.4.2.2 and 2.4.2.3 What kind of statistics (F tests, t tests, etc.) for the cluster permutation statistics? Please be specific about "conditional difference" since this could be between Versions or Identities. How many electrodes did you include in the analysis versus if you only included those averaged "sensor clusters", which needs to be clarified? More clarification is needed to describe the methods to avoid any misunderstanding, such as how many time points, how many electrodes, etc.
Response : Details on these points are described in the revised manuscript.
“bilateral occipital sensors (E39 for the right and E35 for the left) within 100 to 140 ms”
“The left and right occipito-temporal clusters each included five electrode locations (E40, E42, E45, E44, E43 for the right cluster, and E27, E28, E30, E31, E32 for the left cluster) following the previous ERP studies on face processing. The following analyses were carried out for averaged ERP waveform across the five electrodes included in each cluster.”
“global field power (GFP) [47] computed across all the 61 channels by cluster-permutation statistics in the same procedure as described above”
“ERP waveforms at occipito-temporal clusters were compared between the self and others’ familiar faces for original, aesthetically enhanced, and degraded versions by cluster-permutation statistics [46].”
“In this procedure, t-value is computed at each time-point by paired t-test between the ERP waveforms of the two conditions (Identity; Self vs Familiar), and time-points are determined at which the difference in ERP amplitude between self and familiar other’s faces reaches significance threshold. Then, temporal cluster is formed by joining contiguous significant time-points, and the test statistics is computed as the sum of t-values within the cluster. The test-statistics is computed for 1,000 times by random permutation, i.e. randomly shuffling the conditions, through which one can obtain null distribution of test-statistics. The probability of obtaining the observed test-statistics is computed based on the null distribution obtained by random permutation. Maris and Oostenveld [46] have shown that extracting significant temporal clusters by this procedure allows sensitive detection of time-windows of interest while controlling the family-wise error rate to expected level. The family-wise error rate”
“Because the data was down-sampled to 250Hz, each epoch contained 200 time points.”
“In subject-level analysis, data was first down-sampled to 50 Hz (resulting in 40 time points in each trial).”
“Thus, in contrast to the case of diagonal decoding, AUC is computed for every combination of temporal points resulting in a matrix of 40 x 40 AUCs that is generally termed ‘temporal generalization matrix’.”
Comment 8:
- in 2.4.2.3 what kind of algorithm was used to control for multiple comparisons? Since you ran tests over multiple time points this needs to be clarified for cluster-based analysis. What is condition A vs condition B, Identity or Version? Also, It is unclear to me why these particular machine learning methods were chosen given that there are so many different methods available. Please provide ratione to help readers understand better.
Response: Significance level for multiple comparisons was adjusted by the algorithm proposed by Maris and Oostenvald [46] in all the cluster-analyses in the present study. The reason why LDA was chosen, are described in the manuscript.
“Significant temporal cluster was searched for while controlling the family-wise error rate by the permutation clustering approach with 1,000 iterations [46].”
“In the group-level analysis, cluster of AUCs that is significantly different from chance level (AUC = 0.5) was searched for within temporal generalization matrix by cluster-permutation statistics. In contrast to the cluster-permutation statistics described thus far, the cluster in temporal generalization matrix is two-dimensional. Aside from this point, the principle underlying this procedure and the test-statistics are the same with cluster-permutation tests for time-series data [46]. ”
“LDC was chosen because it showed superior performance in classification of ERP data compared to other classification algorithms in previous studies [33, 34]. ”
Comment 9:
Results:
-Several figures have very tiny fonts and are unintelligible in print (Figure 2, Figure 5 5.). Please adjust.
Response: I corrected the figures to improve the intelligibility. Please see also pour responses to Comment 14 by Reviewer #1.
Comment 10:
Conclusions:
-The self vs other in degraded condition shows this later effect around 600ms post stimulus onset, which seems to occur after response (around 492-513 in average). Therefore it seems that this effect does not really entail necessary neural activity for self vs other face processing? How would you explain that?
Response: Thank you for raising an important point. As you pointed out, the fact that self vs other effect emerged after response selection indicates that this effect does not reflect identity (self vs other) discrimination. After identity discrimination, appraisal of self/other visual representation ensues. In my interpretation, this later appraisal process is reflected in the self vs other effects in the degraded condition. This interpretation accords with findings of many ERP studies that effects in long latency range reflects emotional/appraisal process of sensory stimulus processing. In explained this logic to more details in the Discussion.
“Averaged RT for making manual responses was about 500 ms in the present task. Thus, the long-latency effect revealed by multivariate analysis is not directly linked to perceptual discrimination of facial identity, but presumably to later cognitive and evaluative stages of self face processing. Temporal generalization analysis revealed that this effect was temporally stable for the original, but not for the degraded version of faces, which indicates that neural activation underlying the long-latency effect differs between original and degraded versions. ”
Comment 11:
-could aesthetically degrade versions be explained by more masculine features? Then the results would suggest processing between masculine (degraded) versus feminine (enhanced) features of self vs other's faces.
Response: Thank you again for your insightful comment. Faces in the degraded version in the present study exhibit more masculine and more mature morphological features. So, it could be possible that degraded vs enhanced manipulation in the present study is confounded by masculine/feminine and babyish/mature characteristics. This poses some limitation to the interpretation of the data, which I mentioned in the Discussion.
“Lastly and most importantly, findings of the present study tell little about functional significance and psychological mechanisms underlying the observed effects of aesthetic manipulation and facial identity on ERPs. For example, manipulation of facial morphology adopted in the present study probably influenced multiple aspects of facial impression other than facial beauty; large eyes and small nose in aesthetically enhanced version are generally linked to baby-schema [66, 67], and larger jaw in degraded version is a prominent feature of facial masculinity [68, 69]. However, it is unclear which dimension of facial evaluation, e.g. facial babyishness or masculinity, the observed pattern of ERP responses is linked to. ”
Comment 12:
-Only females were included and there is no clear rationale. The results can hardly be generalizable to those who are not young females.
Response:
The data had been collected about 10 years ago. Back then, Japanese females were generally more keenly self-conscious about their appearances than males (things have changed abit since then). This is why I focused on females’ responses. I made this point clearer in the revised manuscript and discussed the limited generalizability of the present findings.
“Only females with relatively small sample size were recruited as participants because previous studies have pointed out that females tend to be self-conscious of their physical appearance [16, 17].”
“First, only young-adult females were recruited as participants, which severely limits the generalizability of the present findings. A recent study [64] reported that larger proportion of females makes effort of editing their selfies shared on social-networking services. Thus, it is highly conceivable that males, who are less accustomed to viewing aesthetically-manipulated self representation, show neural activation different from females.”

Reviewer 3 Report
Summary
In the present study, Doi analyzed the neural mechanisms underlying the processing of actual and deformed (enhanced or degraded) images of self and others’ familiar faces by conventional event-related potentials analysis and multivariate analysis. Their results showed that multivariate analysis revealed differences in ERPs in the long latency range to self and familiar others’ faces only when they were aesthetically degraded. In contrast, conventional ERP analysis (peak and global field power) found no effect. The paper is a very interesting piece of work.
Broad comments
Although the sample size is very low, this work has notable strengths such as the novelty of having found differential pattern of neural activation in response to one’s own and familiar other’s face and that these differences were prominent for aesthetically degraded image of self-face. In addition, it shows new data on the importance of using machine learning-based multivariate analysis in brain processing in general, measured with EEG, and in face processing in particular. Finally, the methodology and analysis performed are well justified, and the main conclusions reached are well sustained by the obtained results. There are no major concerns.
Specific comments
Although there are no major issues, there are a number of minor comments that needs to be addressed before final acceptance:
11. In EEG measurement section, the number of channels/sensors registered and the used device must be specified.
22. In the Figure 2, “Enhanced” appears twice instead of “Degraded”. Please correct.
33. The quality of Figure 5 should be improved, especially with regard to the Y-axes, as the legend and its units are not visible.
44. There is no limitations section. At the very least, it is necessary to point on the low sample size (N=18) which implies that the results and conclusions reached should be taken with caution.
55. In my opinion, it is necessary to include in the conclusion that the differences obtained and discussed are mainly in the higher latencies (late components, 589-682 ms) of the ERPs, and their significance in terms of face processing.
Author Response
Response to Reviewer #3
Below I summarized my responses the comments by reviewer #3. The corrected sentences are highlighted in red in the manuscript.
Please note that the overall results have changed to some extent (especially those of conventional ERP analysis), after I re-analyzed the data by fixing the problem of rank deficiency in ICA and adopting mean amplitude approach instead of peak amplitude/latency analysis following your comments. Consequently, I ended up renovating Results and Discussion section. I sincerely apologize for the confusion.
Comment 1:
- In EEG measurement section, the number of channels/sensors registered and the used device must be specified.
- In the Figure 2, “Enhanced” appears twice instead of “Degraded”. Please correct.
- The quality of Figure 5 should be improved, especially with regard to the Y-axes, as the legend and its units are not visible.
Response:
Details are described on these points and figure were corrected to increase intelligibility accordingly.
“. EEG referenced to Cz was recorded by Geodesic 64ch EEG System (NetAmps 300, Electrical Geodesics Inc, Eugene, Oregon, USA)”
Comment 2:
- There is no limitations section. At the very least, it is necessary to point on the low sample size (N=18) which implies that the results and conclusions reached should be taken with caution.
Response:
Thank you for raising an important point. I mentioned relatively small sample size as limitation of the present study together with other drawbacks of the present study.
“First, only young-adult females with relatively small sample size were recruited as participants,”
Comment 3:
- In my opinion, it is necessary to include in the conclusion that the differences obtained and discussed are mainly in the higher latencies (late components, 589-682 ms) of the ERPs, and their significance in terms of face processing.
Response:
After re-analysis, I found Identity effect in early latency range in conventional ERP amplitude analysis. Only for degraded version, effect of Identity emerged both in early and late latency range. I Included this point in the conclusion. At the same time, the functional and psychological significance of the observed effect remains elusive at this point. I mentioned this limitation in Discussion.
“Effects of identity was observed both in N170 and long-latency range for aesthetically degraded version of faces. Importantly, the effect of self-relatedness in long-latency range was detected only by multivariate analysis based on machine learning but not by conventional ERP amplitude or GFP analysis.”
“Lastly and most importantly, findings of the present study tell little about functional significance and psychological mechanisms underlying the observed effects of aesthetic manipulation and facial identity on ERPs.”

Round 2
Reviewer 2 Report
I would like to commend the author for making substantial revisions and the manuscript is substantially better and clearer. I still have a few more comments and questions:
The electrode notations (e.g., E62, E63) do not conform to the international standardized/conventional terminology (such as PO7, PO8). Please either add a figure with the sensor diagram to represent the electrodes chosen for analysis, or simply add/use standardized notations for better communication of methods.
Please state explicitly what baseline range was used for baseline correction (-100 to 0 ms, or -50 to 0 ms).
You may need to change the headings of 3.2.1 & 3.2.2 from "peak" analysis to perhaps "mean amplitude" analysis, since this part has been revised.
The epoched EEG data that was entered for multivariate analysis (2.4.2.3) and temporal generalization analysis (2.4.2.4). Was the data preprocessed the same way as the ERP data? Also, I still could not find what electrodes were included for these analyses and whether the electrodes were comparable to those used for the conventional analyses. If these were different between conventional vs advanced analyses, then would it be possible that the differences in results could be simply due to how the entry data was manipulated and not due to principles of different analytical approaches?
I see now that you revised the approach of your conventional ERP analysis and you actually found significant results at N170 between self vs others in the degraded version. What's interesting is that the more data-driven approach using cluster-based permutation tests did not identify this difference at this earlier time window. Also, not from the multivariate analysis. So maybe conventional analysis is still good for some? Perhaps it's the poorer temporal resolution in your advanced analysis? Any other thoughts on why? This may be inherently tied to the pros and cons of each method that you should consider elaborating more in the discussion section. Currently you focus only on the potential pros of advanced analysis. Elaborating on this may further strengthen the central message of your study and provide broader implications on what methods should one choose when examining EEG signal.
Line 423 a typo -- "Out" observation should be "Our"
Author Response
Response to Reviewer #2
Below I summarized my responses the comments by reviewer #2. The corrected sentences are highlighted in red in the manuscript.
Comment 1: Please either add a figure with the sensor diagram to represent the electrodes chosen for analysis, or simply add/use standardized notations for better communication of methods.
Response: Some sensor locations included in the electrode cluster analyzed do not correspond to those specified in international 10-20 system. So, following your advise, I included channel layout in Figure 1. Please also note that almost the same sets of electrode cluster were used as occipito-temporal clusters in the previous studies on single-trial analysis of ERP induced by facial stimuli (Tian et al, 2018; Smith et al, 2012)
“Right Panel: Representation of scalp distribution of channels.”
“The channel layout on scalp surface is shown in Figure 1. ”
Comment 2: Please state explicitly what baseline range was used for baseline correction (-100 to 0 ms, or -50 to 0 ms).
You may need to change the headings of 3.2.1 & 3.2.2 from "peak" analysis to perhaps "mean amplitude" analysis, since this part has been revised.
The epoched EEG data that was entered for multivariate analysis (2.4.2.3) and temporal generalization analysis (2.4.2.4). Was the data preprocessed the same way as the ERP data?
Line 423 a typo -- "Out" observation should be "Our"
Response: Thank you for your careful reading. I added these pieces of information and made necessary corrections to the manuscript accordingly.
“with -100 ms to 0 ms as the baseline.”
“The same set of epoched data from 61ch as used in the conventional ERP mean amplitude analysis was used for multivariate analysis.”
“The same set of epoched data from 61ch as used in the multivariate analysis was used for temporal generalization analysis.”
“Our observation of larger P250 amplitude”
Comment 3: Also, I still could not find what electrodes were included for these analyses and whether the electrodes were comparable to those used for the conventional analyses. If these were different between conventional vs advanced analyses, then would it be possible that the differences in results could be simply due to how the entry data was manipulated and not due to principles of different analytical approaches?
………
Also, not from the multivariate analysis. So maybe conventional analysis is still good for some? Perhaps it's the poorer temporal resolution in your advanced analysis?
Response: Different set of electrodes were used in multivariate analysis from ERP mean amplitude analysis (data from all the 61 channels in multivariate analysis and those only from occipital electrodes in ERP mean amplitude analysis). In addition, the data was down-sampled to 50Hz in multivariate analysis. Thus, as you pointed out, it is quite conceivable that these differences in datasets explain differences between the results of conventional analysis and multivariate analysis.
To check this point, I ran multivariate analysis under the same condition as ERP mean amplitude analysis. That is, multivariate analysis was carried out on the dataset from occipito-temporal electrodes sampled at 250 Hz. Multivariate analysis failed to detect the identity effect on N170 observed in ERP mean amplitude analysis, which indicates that the differences between the analytic methods are not attributable to differences in electrodes included and sampling rate at least in the present case. I described the results of this additional analysis in Discussion section.
“Multivariate analysis and cluster-based permutation statistics of ERP time-series failed to detect statistically significant effect of facial identity in N170 component, which indicates that theory-driven approach of conventional analysis still has an edge over data-driven approach in the analysis of early components. The sampling rate of dataset was only 50 Hz and the data from all the 61 channels were used in the multivariate analysis. In contrast to this, only data from occipto-temporal clusters sampled at 250 Hz were submitted to ERP mean amplitude analysis. One may conceive that such differences in datasets explain the failure of multivariate analysis to find any effects in early component. However, it is not the case at least in the presents study. Multivariate analysis of the data in occipito-temporal clusters sampled at 250 Hz still failed to find statistically significant effect in N170 component as observed in ERP mean amplitude analysis.”
Comment 4: What's interesting is that the more data-driven approach using cluster-based permutation tests did not identify this difference at this earlier time window. Also, not from the multivariate analysis. So maybe conventional analysis is still good for some? Perhaps it's the poorer temporal resolution in your advanced analysis? Any other thoughts on why? This may be inherently tied to the pros and cons of each method that you should consider elaborating more in the discussion section. Currently you focus only on the potential pros of advanced analysis. Elaborating on this may further strengthen the central message of your study and provide broader implications on what methods should one choose when examining EEG signal.
Response: Thank you for raising an important point. The failure of multivariate analysis and permutation statistics to find ephemeral effect in N170 component is partly attributable to reduced sensitivity of permutation-statistics to small temporal cluster as pointed out in the previous studies. Conventional mean amplitude analysis also had some advantages over multivariate analysis in a-priori selection of optimal sensor clusters for detecting experimental effects on early components. I described these potential reasons in Discussion.
The overall pattern strongly indicates the necessity of using both data- and theory-driven approaches in a complementary manner to investigate the electrophysiological responses to facial stimuli in its entirety. I emphasized this point in the conclusion section.
“Occipito-temporal electrodes were chosen based on the existing findings [19, 20, 62] to optimize the chance of finding any effects of experimental manipulation on early visual components. This is one advantage of ERP mean amplitude approach over multivariate analysis. Another reason for superior performance of ERP mean amplitude analysis is its sensitivity to short-lasting effect. Previous study [34] have pointed out the possibility that cluster-based permutation is less sensitive than alternative procedures such as FDR correction [63] in detecting small cluster of activation, which is disadvantageous particularly for detection of ephemeral effect in early latency range. It is of great value for advancement of data-driven analysis of ERP data to refine novel procedure to balance sensitivity to and controlling false-positives of meaningful but short-lasting temporal clusters.”
“In contract, identity effect on N170 component was observed only by conventional approach of ERP mean amplitude analysis. Taken together, these results suggest the necessity of integrating both theory- and data-driven approaches in a complementary manner”
In addition to incorporating your comments, I corrected several minor errors/typos throughout the manuscript. During this process, I found out an error in randomization process in analysis script. So, I re-run the multivariate analysis, and updated the results. The results and conclusion remain essentially unchanged.
I thoroughly checked the entire analysis scripts again (especially those I newly wrote for the first round of revision), and confirmed that everything is in order.
